

# Intraspecific variation in the pterosaur *Rhamphorhynchus muensteri*— implications for flight and socio-sexual signaling

Michael B. Habib[1,2] and David WE. Hone[3]

[1] Dinosaur Institute, Natural History Museum of Los Angeles County, Los Angeles, California, United States
[2] Medicine, University of California, Los Angeles, Los Angeles, CA, United States
[3] Queen Mary University of London, London, United Kingdom

## ABSTRACT

Pterosaurs were the first powered flying vertebrates, with a fossil record that stretches back to about 230 million years before present. Most species are only known from one to three specimens, which are most often fragmentary. However, *Rhamphorhynchus muensteri* is known from numerous excellent specimens, including multiple specimens with soft tissue preservation. As such, *Rhamphorhynchus muensteri* is one of the only pterosaurs amenable to analysis for intraspecific variation. It has been previously predicted that elements directly involved in the flight apparatus, such as those of the forelimb, will be more highly constrained in their proportions than other parts of the skeleton. We investigated the degree of variation seen in elements and body parts of *Rhamphorhynchus*, which represents the best model system among pterosaurs for testing these expectations of intraspecific variation. We recover evidence for high levels of constraint throughout the appendicular and axial elements (head, neck, torso, tail, forelimbs, hindlimbs), suggesting that all were important for flight. We further find that tail variation increases among the largest specimens, suggesting reduced constraint and/or stronger sexual selection on the tail in more mature individuals.

## INTRODUCTION

*Rhamphorhynchus muensteri* is from the Late Jurassic limestone beds of Southern Germany and is one of the largest non-pterodactyloid pterosaurs (*Wellnhofer, 1975*). It is extremely well known as a result of numerous specimens having been discovered and described (*Wellnhofer, 1975*). These specimens cover a wide range of sizes and ontogenetic states (*Bennett, 1995*) and include both young juveniles and unusually large adults, covering a near order of magnitude in wingspan with animals from 0.29–1.72 m in wingspan (*Hone et al., 2020*). Like all pterosaurs, *Rhamphorhynchus* was capable of powered flight, though was potentially unusual in that young juveniles were precocial and flew from a very young age (*Bennett, 1995*; *Prondvai et al., 2012*; *Hone et al., 2020*).

Corresponding author
Michael B. Habib,
mbhabib@mednet.ucla.edu

As is the case with many konservat lagerstaetten deposits, many specimens of *Rhamphorhynchus* are crushed and visible in only two dimensions. However, specimens of *Rhamphorhynchus* do include those with exceptional preservation, including specimens in three dimensions and/or with extensive soft tissues preserved. Soft tissues preserved in *Rhamphorhynchus* specimens include wing membranes, tail vanes and beaks (*Frey et al., 2003*; *Witmer et al., 2003*; *Hone, Habib & Lamanna, 2013*; *Hone et al., 2015*). *Rhamphorhynchus* was primarily piscivorous (*Wellnhofer, 1975*) although there is evidence of a more varied diet that may have included other vertebrates and invertebrates such as cephalopods (*Hone et al., 2015*; *Hoffmann et al., 2020*). There are also indications that diet may have varied with ontogeny (*Bestwick et al., 2020*)

*Rhamphorhynchus* should be regarded as a model organism for pterosaurs as it is one of, if not the, best-studied of pterosaurs and is represented by an extremely good collection of specimens. *Pteranodon* by contrast is known from far more numerous specimens, but these are typically very incomplete, near universally crushed, and include few animals that are not near or at adult size (*Bennett, 2001*). Similarly, some recent discoveries of mass mortality sites for pterosaurs present numerous extremely-well preserved animals (*e.g.*, *Manzig et al., 2014*) but have yet to be described or illustrated in great detail and are often disarticulated. However, new and well-preserved specimens of *Rhamphorhynchus* are still being discovered, entering public collections and being described (*e.g.*, *Frey & Tischlinger, 2012*; *Hone et al., 2015*) and other specimens are being rediscovered and entering the scientific literature (*Ősi & Prondvai, 2009*) so that this resource is still growing. Numerous papers have been published collectively describing its anatomy in great detail (*e.g.*, *Wellnhofer, 1975*, *Bonde & Christiansen, 2003*; *Frey et al., 2003*; *Witmer et al., 2003*; *Bennett, 2015*; *Bonde & Leal, 2015*). It has formed the basis or a major part of important pterosaur studies on growth and ontogeny (*Bennett, 1995*; *Prondvai et al., 2012*; *Hone & Mallon, 2017*; *Hone et al., 2020*), pneumaticity (*Bonde & Leal, 2015*), sexual selection (*O'Brien et al., 2018*), brain structure (*Witmer et al., 2003*), and wing structure (*Bennett, 2015*) and other soft tissues (*Beardmore, Lawlor & Hone, 2017*). Coupled with the high number of well-preserved and often complete (if crushed) specimens, this makes *Rhamphorhynchus* an unsurpassed resource for testing hypotheses about pterosaur biology.

Through its ontogeny, *Rhamphorhynchus* was highly isometric (*Hone et al., 2020*) presenting a remarkably consistent growth trajectory across a four-fold range of length and over a hundred specimens. However, the extent of the intraspecific variation across this collection of specimens has not been assessed. In particular, as flying animals, the constraints of powered flight might be expected to have imposed limitations in the variation of most skeletal elements producing generally little variation. On the other hand, socio-sexual signaling has been suggested for the soft tissue tail vane (*O'Brien et al., 2018*) suggesting that the tail may have been under selection for signaling and less for flight (though both may still be important). If so, the tail would be expected to be more variable than other elements. Traits under strong sexual selection are known to show high rates of intraspecific variation compared to other morphological traits (*Fitzpatrick, 1997*; *Emlen et al., 2012*), including the tails of flying animals (*Alatalo, Höglund & Lundberg, 1988*).

However, proportions related to flight performance are heavily constrained and are known to show relatively low rates of intraspecific variation (*Dyke, Nudds & Rayner, 2006*).

These observed patterns of variation agree with predictions from functional constraint theory as applied to flying animals. Relatively small variations can have disproportionate effects on flight performance, given that the large forces and specific requirements of lift generation in flight necessarily constrains the morphology of flying animals in numerous ways (*Habib, 2013*). In pterosaurs, this included constraints on wing position (*Palmer & Dyke, 2012*) and wingtip material properties (*Hone et al., 2015*). Given these constraints, as well as the measured rates of intraspecific variation in living flyers, one would expect that rates of intraspecific variation in functionally-relevant characters in pterosaurs might be relatively low. By contrast, given that sexually-selected characters show high rates of intraspecific variation, structures such as crests and tail vanes are likely to be highly variable between individuals (*Hone, Naish & Cuthill, 2012*).

Here we investigate the degree of variation seen in elements and body parts of a pterosaur. *Rhamphorhynchus* represents an excellent taxon for testing these expectations of intraspecific variation, owing to the large number of relatively complete specimens, thorough descriptions, the presence of traits hypothesized to be primarily under mechanical performance selection (*i.e.*, wing elements), and the presence of traits thought to be sexually-selected (*i.e.*, tail vanes).

## METHODS

We took the recently compiled dataset of *Hone et al. (2020)*, which contains measurements for 137 specimens of *Rhamphorhynchus*. This is itself an extension of the dataset of *Wellnhofer (1977)* and as a result, follows his breakdown of skeletal elements and collections of elements (Table 1). We follow *Bennett (1995)* in considering all specimens of *Rhamphorhynchus* to belong to a single species, *R. muensteri*, and therefore, we consider our analysis to be intraspecific. However, we do acknowledge the reservations articulated by *Bonde & Leal (2015)* in this regard, and note that our analysis may, instead, be intrageneric. In consideration of future taxonomic changes (and other factors), we have included the full list of specimens, with all measurements, as a Supplemental File (See Supplemental Information). Furthermore, we considered, *post-hoc*, a specimen that has been assigned to a second species of *Rhamphorhynchus* (*Rhamphorhynchus etchesi*).

We tested multiple hypotheses regarding element variability in *Rhamphorhynchus*. First, because wing elements are known to be highly constrained in flying animals relative to closely related non-flying species (*Dyke, Nudds & Rayner, 2006*; *Pennycuick, 2008*), we hypothesized that the limb elements would show lower intraspecific variation than other elements in the skeleton. In pterosaurs, the hindlimbs are integral with the wing membrane (*Elgin, Hone & Frey, 2011*) and therefore we would predict that the femur and tibia would show similar constraints to the forelimb elements. Second, because the tail in *Rhamphorhynchus* includes a soft tissue "vane" at the tip that shows a growth pattern consistent with it being a socio-sexual signal, we hypothesized that relative tail length would be more variable than other proportions in *Rhamphorhynchus*. Third, we hypothesized that tail length variability would be greater in larger, putatively mature

**Table 1 Summary of descriptive statistics for intraspecific variation in *Rhamphorhynchus muensteri*.**

| X var | Y var | Avg std res | Res compar (p) |
|---|---|---|---|
| MC IV | Tail | 0.72 | **0.002** |
| MC IV | WP1 | 0.57 | 0.074 |
| MC IV | PCRW | 0.84 | **0.002** |
| MC IV | Femur | 0.77 | 0.061 |
| MC IV | Skull | 0.74 | 0.128 |
| MC IV | WP2 | 0.67 | **0.005** |
| MC IV | Humerus | 0.28 | 0.051 |

Note:
Statistically distinguishable mean comparisons ("significant") are designated in bold.

individuals, as these individuals would be those engaged in signaling under the socio-sexual selection model of tail vane ontogeny evolution.

Basic descriptive statistics were generated for all 14 variables, including standard deviation, relative standard deviation, standard error of the mean, mean, median, and mean-median differential (as a basic measure of skew).

For the purposes of having a standard variable against which to measure relative variation in other traits (designated as the only X variable, with all others designated as Y1 through Y13), we primarily used the length of metacarpal (MC) IV. We note that, as per *Hone et al. (2020)*, the fourth metacarpal displays negative allometry, while most of the elements scale isometrically. We used MC IV as our standardized X variable, despite negative allometry, because MC IV had lower relative variability (measured simply as standard deviation/mean) than the other traits (except for orbit size which has too small a sample to use as the standard of comparison). In order to confirm that the allometry in MC IV did not affect results, we also ran metrics of tail variation (the most important variable for our hypotheses) against variables that scale isometrically. Furthermore, most of our analyses relied on the use of residuals from scaling regressions, rather than the slopes of the regressions. We confirmed that residuals were not patterned with respect to the X values in regressions (*e.g.*, residuals were properly standardized to slope of zero). The residual values are not affected by the slopes of the underlying regressions, so long as they are standardized properly, since the residuals only measure the distance (with sign) from the value predicted by regression.

When residuals were plotted, there were two data clouds apparent: one for individuals with MC IV lengths under 15 mm and another for MC IV lengths over 16 mm. These size groups correspond closely to the split between the small and large classes found by *Bennett (1995)*. These larger specimens, which have a skull length over 80 mm and humeral length of over 30 mm, have a fused scapulocoracoid and completed fused pelvis suggesting they are close to, or at, adult size in this regard. This size class also corresponds to the point where the tail vane undergoes accelerated growth such that it is growing faster than the isometric humerus (*O'Brien et al., 2018*) with the individuals of *Rhamphorhycnhus* before this point having a juvenile tail vane morphology, and those after the adult form. We binned those individuals with MC IV lengths under 15 mm as "small" individuals

(putatively immature) and those individuals with MC IV lengths over 15 mm as "large" individuals (putatively mature). In all cases, we refer to maturity as osteologically maturity. However, this is known to correlate strongly with size and putative sexual maturity in close relatives (dinosaurs) (*Hone, Farke & Wedel, 2016*). We then compared mean standardized residuals between these two groups for the hypothesized low constraint (tail) variable and a hypothesized high constraint variable (WP1).

As a simple method for determining if any variable pairs were uncorrelated (which would be surprising) we ran a series of Pearson product moments between all pairs of variables. We did not generate *p* value calculations for the product moments, because the permutation effects would have rendered any alpha values worthless in this case (*i.e.*, the required Bonferroni Correction is too large to be practical). Instead, potential outlier relationships would have been further tested with a glm *post-hoc* test, but no such potential outliers were detected. All correlation coefficients were large and broadly similar. We therefore proceeded with the remaining analysis with the working expectation that all measured traits are autocorrelated on the basis of size.

We examined the fifth pedal digit and its variation across specimens, but it was excluded from our statistical analyses because of a probable preservation bias (see Discussion). The fifth toe appears to be hyper-variable due to warping post-mortem, rendering it useless for our study.

## RESULTS

Contrary to our first hypothesis, limb elements did not show lower intraspecific variation than other elements in the skeleton. No specific osteological variable was found to be outstanding with regards to its variability or correlation with other proportions (Fig. 1). Limb elements in *Rhamphorhynchus* were not found to be less variable, on average, than other portions of the skeleton. Contrary to our second hypothesis, relative tail length was not more variable than other proportions in *Rhamphorhynchus*. All proportions appear to have similar levels of constraint. Tail length was not found to be any more variable than other proportions, taken over the full set of specimens. However, results did provide some support for our third hypothesis, regarding an increase in tail length variation among larger individuals. Standardized residuals in tail length when regressed on metacarpal IV length show a statistically significant increase between the small and large size groups ($p < 0.002$). Residuals in the length of most other elements, regressed on the same set of metacarpal lengths and divided into the same size groups do not show a significant difference in mean value (Table 1) (Statistically distinguishable mean comparisons ("significant") are designated in bold in Table 1). However, variation in the combined length of the trunk vertebrae also show a statistical disparity between large and small individuals, as does one wing element (WP2). Some of this disparity may be a result of comparatively higher precision, in a relative sense, of measuring to the same absolute level of accuracy on larger elements. However, proper standardization of the residuals in each of the variables utilized suggests that this measurement difference has a small, non-significant effect.

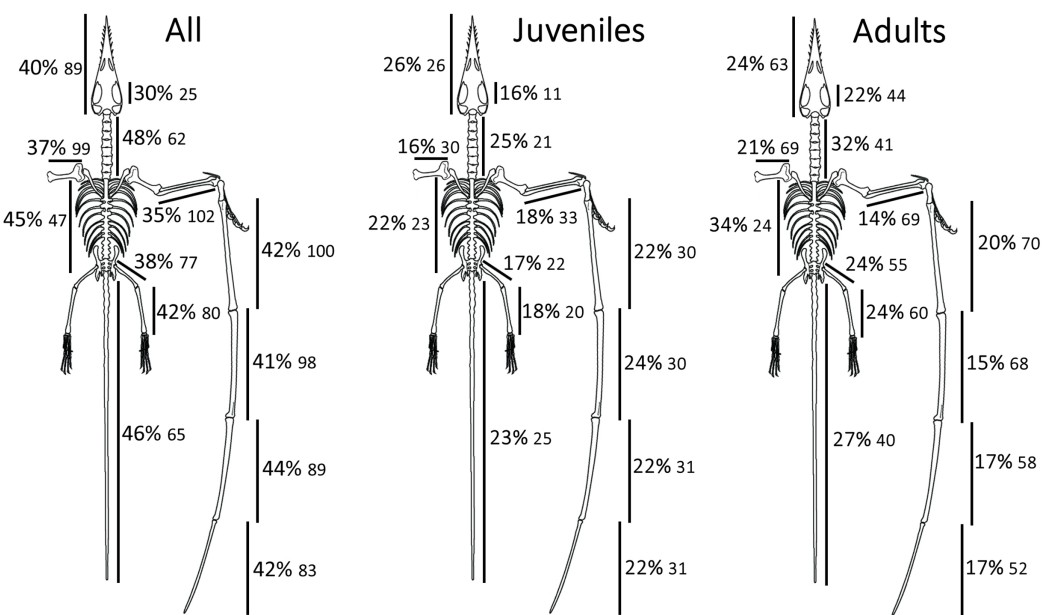

**Figure 1 Summary of relative rates of variation by region or element.**

While we did not run a dedicated effect size test, we noted that the variability in WP1 and WP2 residuals among the "mature" (large) size group seemed to be disproportionately driven by just two individuals with particularly short wing phalanx elements relative to MC IV, based on visual examination of the residual plots. However, removing these two individuals, generating a new regression, and re-standardizing residuals did not change the overall conclusions (recovered *p* values both increased substantially from a pure numerical standpoint, but WP2 variation in adults remained significantly different from juveniles. WP1 variation differential remained non-significant).

To make sure that the differences were not driven by a small number of specimens whose impact was not obvious from qualitative assessment of residual plots, each comparison was put through a resampling series in which two specimens were removed at random and the results were re-calculated. This was done 100 times for each comparison. None of the results changed in their statistical significance under this resampling.

Descriptive statistics returned one outlier, namely the proxy for skew in total length of the trunk vertebrae (PCRW). The distribution of PCRW was found to be highly "right" skewed, while all other variables have a modest "left" skew (Table 1). Rates of linear dimension variation are all highly consistent across the skeleton. If sampling effects on standard deviation are taken into account, the relative variability in each trait becomes nearly identical. This might be indicative of a highly constrained system.

All residuals were found to be unpatterned with regards to the X values of their source regressions (*i.e.*, all were properly standardized), such that the residuals of different variables could be cross-compared as a more thorough, scaling-independent measure of variation. The average of magnitude of residuals were consistent across variables, further

demonstrating that no particular variable in the sample shows increased rates of intraspecific variation relative to the others (see Supplemental Data).

Comparisons of *Rhamphorhynchus etchesi* lengths and length ratios with all other *Rhamphorhynchus* revealed overall large size and elongate distal wing elements in *R. etchesi*. Its proportions are particularly distinct with regard to the ratio of MC IV to WP2, even in the context of the substantial intraspecific variation recovered for this ratio (see Table 1). The entire distal wing of *R. etchesi* was found to be considerably elongated when compared to other *Rhamphorhynchus* (see Supplemental Information).

## DISCUSSION

*Rhamphorhynchus* is known from multiple localities that represent at least 4 (likely over five) million years (*Bennett, 1995*), but there are no good data to suggest that any of the specimens or sets of these are anomalous in their anatomy, size or distribution. Instead, these specimens seem to represent genuine sampling of a species across an extended period of time. Unfortunately, the records for many specimens are old enough that their original localities and/or formations are not recorded. As a result, sample sizes for any given age are too small to be meaningful. Therefore, we could not look at variability over time, only over size.

Tail length was not found to be any more variable than other proportions in a general sense, which is indicative of a functional constraint on tail length. However, it is notable that there is a significant increase in relative variability in tail length among larger, putatively mature individuals. Combined with prior work that demonstrates high rates of variability in the tail vane through ontogeny and between individuals (*O'Brien et al., 2018*), this sudden "jump" in tail length variability is consistent with the hypothesis that the tail took on a signalling function in adults. Our results strongly support a signalling function for the tail in adult *Rhamphorhynchus*, particularly for the tail vane. Combining our results on tail length variation with prior work on the tail vane variation, we can now conclude that the vane represented a focus of variability attached to a low-variability structure, and this variation increased significantly in larger animals at the point that the vane grew. This is indicative of a signaling function. At the same time, the similar jumps in variation among the trunk vertebrae lengths and length of the second wing phalanx suggest that adults might have been somewhat more variable, overall, in their proportions. This might be the result of differences in environmental impacts, nutrition, and other growth-impacting factors over the comparatively longer lifespans of the more mature individuals. Our conclusions in this regard must necessarily be tempered by our use of a fairly simple, univariate approach incorporating only element lengths. While the simplicity of our approach has the advantage that it can incorporate data from a large sample of specimens (that vary in details of preservation), this streamlined approach also precludes more detailed conclusions regarding the mechanics, growth, and variability of tail vanes. This is particularly true given that a minority of specimens preserve the soft tissue vanes.

If sampling effects on standard deviation are taken into account, then the relative variability in each trait becomes nearly identical. This might be indicative of a highly constrained system, overall, and is consistent with indicators that *Rhamphorhynchus* was

fully mobile (and likely even volant) from an early age (*Hone et al., 2020*). The constraints on tail length in immature individuals suggest that tails were mechanically important in some respect and impacted overall functional performance. The pterosaurian tail has been implicated as having potential for some manner of general control and/or stabilization functions in flight (*Bennett, 1995*; *Chatterjee & Templin, 2004*), though without reference to specific control functions. *Wellnhofer (1991*, p52) and *Unwin & Martill (2003)* specifically suggest that the tail may have functioned in yaw control and "ruddering".

The vertical orientation of the tail vane in *Rhamphorhynchus* (*Wellnhofer, 1978*) does give the appearance of a yaw stabilizer, and these earlier works based their conclusions on the reasonable expectation that pterosaurs were using span-wise lift distributions similar to aircraft (elliptical). However, it is now known that flying animals use an alternative lift distribution (gaussian) and, therefore, have no use for vertical, yaw-correcting rudder systems, since their wings produce proverse, rather than adverse, yaw when turning (*Bowers et al., 2016*). Notably, this explains the lack of vertical stabilizers in living flying animals: the vertical tails of aircraft serve to counter adverse yaw produced by the wingtips, as a side effect of their elliptical lift distribution. With their alternative lift distribution, flying animals do not incur this adverse yaw and need no vertical corrective surface.

Tail fans and vanes at the end of a long, reinforced tail can provide usable pitch authority, and the tail fan in microraptorines has been specifically implicated as providing pitch authority during flight (*Han et al., 2014*). However, pterosaur tail vanes did not have an orientation that would provide notable weight support and/or pitch authority. Furthermore, basal monofenestratans reduced the tail compared to other non-pterodactyloids, and both anurognathids and pterodactyloids reduced it still further (*Lü & Hone, 2012*), demonstrating that a long tail was not critical for flight in pterosaurs.

The tail might have still provided important flight control in *Rhamphorhynchus* through functions other than yaw production, however. Tail motions might have provided some flight control capacity by shifting the center of mass relative to center of lift. In this model, the distally located tail vane might provide some additional effect by adding more mass far from the body. That said, we propose that the tail might have been most important in providing control and balance functions during non-flighted locomotion (*i.e.*, running and/or climbing). This functional explanation is consistent with striking similarities between the tails of dromaeosaurids and rhamphorhynchids, including the relative dorsoventral rigidity of the tail (provided by caudal rods) and the comparative lateral mobility of the tail (*Persons & Currie, 2013*).

It is unclear why there are so few specimens in the "mid-range" size class (*i.e.*, near 15 mm MC IV length). One possibility is that this size range represents a point of accelerated growth. If so, then specimens in this putative rapid growth phase would be under-sampled because they would spend a comparatively short period of time in this size class. This is consistent with the idea that this size "gap" is coincident with the onset of maturity. Another possibility is that behavioral changes over ontogeny created some manner of preservation bias, with mid-sized individuals less likely to die in locations that promoted fossilization. Additional explanations are also possible, and with no strong

evidence of any particular explanation at this time, all remain speculative. We tentatively prefer the rapid growth phase hypothesis at this time.

The ratios between various elements and sections of the limbs are major traits used in the taxonomy and systematics of pterosaurs (*e.g.*, *Kellner, 2003*; *Andres & Qiang, 2008*; *Padian, 2008*; *Frey, Meyer & Tischlinger, 2011*; *Lü et al., 2015*; *Dalla Vecchia, 2019*; *Zhang et al., 2019*). The evidence here that nearly all major elements are highly conserved between animals of very different sizes and across a considerable period of time provides confidence that such measurements are a reliable source of data in this regard. If intraspecific variation is low, then the proportions of single specimens are likely to be representative of a species of a whole and can be used to help identify conspecifics. Similarly, if other pterosaurs are also highly isometric as seen in *Rhamphorhynchus* (*Hone et al., 2020*) then such traits will be accurate in both adults and juveniles. There will still be intraspecific variation and allometry of some traits, and crushing and distortion of compressed specimens will affect their measurements, so subtle differences in proportions are less likely to represent genuine differences, but overall, the use of such ratios in pterosaur taxonomy and systematics is likely to be appropriate. We note that specimen MJML K-1597, previously identified as a new species within *Rhamphorhynchus* as *R. etchesi* (*O'Sullivan & Martill, 2015*) is more proportionally disparate relative to *R. muensteri* than we expected. The proportions of the wing elements fall outside two standard deviation ranges for the ratios between the humerus, radius, fourth metacarpal, and (especially) the wing phalanges. Other proportions within, MJML K-1597 are consistent with other *Rhamphorhynchus*, but the distal part of the wing is more elongate: WP2, 3, and 4 ratios with MC IV all fall outside two standard deviations of these ratios in the general dataset. Given the sample size here and the overall consistency of *R. muensteri*, this suggests that *R. etchesi* is more distinctive than previously considered. We further note that taxonomic choices made with regards to the importance of wing element ratios in MJML K-1597 could also have implications for arguments of taxonomy in other taxa, such as *Pteranodon* (*vs. Dawndraco*)) and Solnhofen ctenochasmatoids.

One element that may have less taxonomic utility than previously thought is the fifth pedal digit, which has also been used to diagnose pterosaurian taxa (*e.g.*, *Wang et al., 2010*; *Lü et al., 2011*; *Zhou & Schoch, 2011*). In non-pterodactyloid pterosaurs, this element has a distinctive angle in it to make the bone a broad V-shape. Given that the fifth toe is integrated with the uropatagium as part of the flight apparatus, it should should the similar high consistency of elements seen here, however it has been noted that this curvature can vary markedly in *Rhamphorhynchus* (*Sullivan et al., 2014*). Although not directly assessed here, the angle in this element does vary considerably (we measured a range of 123–154° between just nine specimens—Fig. 2) and the use of this angle as a taxonomic trait (*e.g.*, *Wang et al., 2010*) is therefore questionable. We suggest that this apparent variation is, at least in part, likely due to differing orientations of the element when it is preserved. As noted by *Bennett (2001)* and as seen with the variation in pterosaurain wingtip curvature (*Hone, van Rooijen & Habib, 2015*), depending on the plane in which crushing occurs, a curved element can reduce, but not increase, its apparent curvature. Therefore, we suggest that the higher angles seen here for the bend in the fifth toe are likely to be more

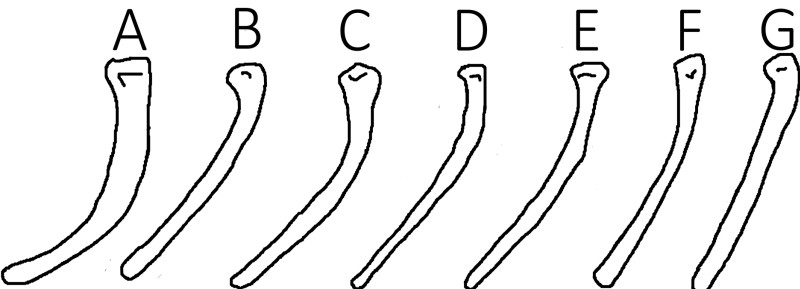

**Figure 2 Variation in toe curvature showing exceptional intraspecific differences likely attributable to post-mortem effects.**

accurate, and that lower values reflect crushing and distortion of an otherwise conserved element.

Additionally, the distal phalanx of the fifth pedal digit may have been comprised of relatively low modulus bone, as has been indicated for the most distal phalanx of the wing finger (*Hone et al., 2015*). While this prospect requires further investigation, it would make sense mechanically, since the fifth pedal digit of non-pterodactyloids was incorporated into the edge of a flight surface in a roughly similar way as the phalanges of the fourth finger. The fifth pedal digit would have been subjected to similar loads as the distal wing finger phalanx on account of membrane tensioning, which would place it under similar mechanically dominated selective regimes. The variation of the angles observed could be due to different loading regimes between individuals, or differences in drying and postmortem tensioning of the membrane (in addition to crushing). These additional sources of non-phylogenetically informative variation make digit V angles poor characters for systematic use.

In this dataset we are unable to compare differences between the left and right sides of individual elements and this would be a worthy future analysis. Variation is seen between left and right elements in many pterosaur specimens although again, there should be caution because of the effects of crushing and distortion in Largerstaetten-type specimens. Such work may therefore be best tested on 3d preserved material.

We cautiously interpret our results as being indicative of a signalling function for the tail of Rhamphorhynchus, with the caveat that some of this variation might be some form of correlated growth variation among the more distal parts of the skeleton, generally. Regardless, we hope that our analyses provide another step towards establishing *Rhamphorhynchus* as a model system for studies of growth, ecology, ontogeny, and functional morphology in extinct vertebrates.

## ACKNOWLEDGEMENTS

We wish to thank the two reviewers for their thoughtful, constructive feedback on our manuscript. We also wish to thank Rebecca Gelernter for graciously allowing us to use her illustrations.

### Funding
The authors received no funding for this work.

### Competing Interests
David WE Hone is an Academic Editor for PeerJ

### Author Contributions
- Michael B. Habib conceived and designed the experiments, performed the experiments, analyzed the data, prepared figures and/or tables, authored or reviewed drafts of the article, and approved the final draft.
- David WE. Hone conceived and designed the experiments, performed the experiments, analyzed the data, prepared figures and/or tables, authored or reviewed drafts of the article, and approved the final draft.

### Data Availability
The measurements and summaries of individual tests with residual standardizations are available in the Supplemental File.

### Supplemental Information
Supplemental information for this article can be found online at http://dx.doi.org/10.7717/peerj.17524#supplemental-information.

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
