# Peer review of "Intraspecific variation in the pterosaur Rhamphorhynchus muensteri—implications for flight and socio-sexual signaling"

_PeerJ, doi:10.7717/peerj.17524_

## Round 0.1 · original submission · Minor Revisions

We have now received two reviews of your work, both concurring that it is worthy of publication. They commend your efforts while suggesting some areas for improvement.

Since Bennett's (1995) work, the Rhamphorhynchus specimens from the Late Jurassic of Germany have typically been regarded as belonging to a single species—a perspective you have also adopted. However, it is necessary to acknowledge the reservations articulated by Bonde and Leal (2015) in this regard.

Several sentences (e.g., line 38) could benefit from reformulation for improved clarity and precision. Additionally, clarification is needed to ascertain whether the mention of 'Hone, Farke, and Wedel' on line 142 refers to a specific paper.
Thank you.

Reviewer 1 ·

Basic reporting

The paper is readable and well structured with sufficient SI to back the claims. Its intent is clear, methodology and results coherent, with discussion not speculating on anything not covered by the study.

There are some minor formatting mistakes. Such as occasional lack of italicization (42, 202, 285, 333) or full stops (142), a full readthrough should patch these up. Some elements are missing, like "represent at least XX million years" (208). The SI material needs a bit of clearing up as it has the author's notes still present, like "I need to finish this matrix"; and some error notes that seemingly have not been corrected. More discrepancies might be dotted around and fished with a quick correction. The introduction also repeats talking points about it being a model studying organism (45-46 and then again 88-89), this can be shortened.

(240) Cite source for vertical orientation
(266) There are other interpretations for the biases of single-size group over-representation that could be briefly mentioned

What is considered an "adult vane form" referencing the Bennett (1995) scale? Would it be 3-4 or 5-6?

Experimental design

I like the methodology for mitigating the effects of a limited sample size in 181-187. And mentioning limits of variable measurements in 173.

While sexual signalling is used throughout the text, there's no attempt to show if tail differences are affected by sex. If vanes are sexually dimorphic, is there are bimodal distribution to overall tail dimensions? Are there any preservation conditions and physiology that might affect representative preservation and representation of certain sexes (as it is in Pteranodon)? Limits of relying on a single variable (relative element length) to derive a conclusion should be discussed and flaws that might invalidate findings, or be unverifiable due to lack of data, should be briefly brought up.

The segments on the fifth finger are plausible, along with the re-assessment of R. etchesi (remember to italicise); can there be an SD plot to show it stands out? (285) Do other "Rhamphorhynchinae" like Nesodactylus also plot outside SD? (optional suggestion)

Validity of the findings

As optional suggestions that'd make the paper's conclusions more robust and the paper more multi-variate (although, it'd take considerable time and is not essential to support the concluded points)

The study is very univariate in its approach, given the tail vane is a multi-dimensional feature that can be extrapolated with more than a relative measurement. Are there any other morphologic changes occurring to vanes in animals of a large size in the accentuation of asymmetry and width? Given the preferential preservation of certain size groups, it would be nice to show how many specimens are per sample size in text (outside of just SI and Figure). The two-size bins (large, and small) now encompass a large range of sizes. The large bin sizes take in a much wider sample size compared to small specimens. According to SI, "small individuals" can have humeri as low as 14 mm to 30 mm as "large individuals" while humeri in large individuals can reach 52, 64 to even 79 mm. More than 100% of the initial "large" size.

Concluding, a robust paper with good methodology limiting effects that would create artificial pulses in the results. To make it stronger I'd optionally suggest adding other variables (tail vane dimensions), and discourse about sexual bimodality (present, not present, why hard to test. was it tested in the past). Highlighting areas of missing understanding and general clean-up in SI & text.

·

Basic reporting

This is a very interesting and elegant study, with well-defined, objective, and relevant hypotheses, and a clear and adequate methodology for testing them. I highly recommend its publication. The paper is very well-written and presented, and I have only some minor suggestions, so that I recommend a “minor revision” only.

Introduction

- Paragraph 2: Please note Bestwick et al. 2020 and ontogenetic variation in diet.

- Paragraph 3: In the sentence about “growth and ontogeny”, please notice Bennett 1995.

- Paragraph 4: “huge range of sizes and large numbers of organisms” is vague. I get the meaning because I’m familiar with the background, but it would be best to rephrase this.

Results

- The sentence “It is a disproportionate contributor to the significant result returned for MC IV vs WP2” seems to imply that the holotype of R. etchesi was included in the analyses. Is this correct? In case this is correct, I am a little bit confused because I could not find this specimen in the Excel file. Also, in case this is correct, what happens if you remove R. etchesi from the calculations? I suggest it should not be included (even if it is congeneric with R. muensteri, it’s still not the same species). Either way, please clarify this in the Materials and Methods.

- I have noticed some personal notes on the Excel file. Is this intentional or did you forget to delete them?

Discussion

- Please correct “XX million years”.

- The results regarding the considerable morphometric discrepancy between R. etchesi and R. muensteri are very interesting. It is interesting that this corroborates the species-level distinction between the two, as well as the taxonomic usefulness of these morphometric values – what is very relevant. However, regarding the suggestion that it “may well represent a separate genus”: why exactly does the reported morphometric distinction provide evidence for generic distinction, rather than simple species distinction within a same genus? Please rewrite this part with more caution, bringing attention to the latter possibility. Optionally, I’d also call attention to the fact that phylogenetic analyses are crucial to support generic assignments, and none was provided when the species was named.

- Regarding the sentence “[…] such as Pteranodon (vs Dawndraco and others) and Solnhofen ctenochasmatids (Pterodactylus, Aerodactylus, and others)”. Please specify “others” and please provide references for the taxonomic issues regarding the taxa in question. Also, please correct “ctenochasmatids” (Pterodactylus and Aerodactylus are not ctenochasmatids, under any sense/phylogeny). I believe you meant “ctenochasmatoids” (sensu Unwin 2003). You could also use “euctenochasmatians” or “archaeopterodactyloids” or something more or less equivalent.

- Please correct “Figure X”. I believe that’d be Figure 3.

- Regarding fifth toe curvature, the suggestion that the lower values are probably due to preservation is very interesting. Have you ruled out allometry? Why was it not included in the statistical analyses? If that’s because of the probable preservational bias, it would be nice to highlight this.

- Regarding the phrase: “Furthermore, both phalanges in question represent the distalmost phalanx of the most embryologically medial digit of their limb segment (the hind limbs become “flipped” by embryological limb rotation).” I am not sure I follow the rationale here. I understand that the fifth toe is the “most medial” toe in an embryological sense because the hindlimbs are flipped. Still, how is the fourth manual digit the “most medial” finger? The fourth manual digit is actually the “most lateral” finger, right? Anyway, why should a “medial most” status imply any relatedness in embryological pattern? “Medial most” status does not imply in serial homology, especially because this status depends on how many digits have been retained, or lost, in each limb. Are you certain this is a good speculation for this paper? Perhaps you could provide further reasoning and elaborate further, but I would rather delete this sentence. It does not change the rest of the reasoning of the paragraph, which is solid enough and convincing enough.

And that’s all! I congratulate the authors on the very interesting paper, and am looking forward to seeing it published!

Best regards,
Rodrigo Pêgas

Experimental design

Excellent.

Validity of the findings

Excellent.

---

## Round 0.2 · accepted · Accept

I confirm that your work is now accepted for publication. However, I would like you to consider adding a "Conclusion" header on line 351, and if you feel it is appropriate, an "Acknowledgements" section as well.